**Kinetically Controlled Glass Transition Measurement of Organic Aerosol**
**Thin Films Using Broadband Dielectric Spectroscopy**
Yue Zhang[1,2,£], Shachi Katira[3], Andrew Lee[1,†], Andrew T. Lambe[2], Timothy B. Onasch[1,2], Wen
Xu[2], William A. Brooks[2], Manjula R. Canagaratna[2], Andrew Freedman[2], John T. Jayne[2], Doug
R. Worsnop[2], Paul Davidovits[1,*], David Chandler[3, §], Charles E. Kolb[2,*]
*1 Department of Chemistry, Boston College, Chestnut Hill, MA, 02459*
*2 Aerodyne Research Inc., Billerica, MA, 01821*
*3 Department of Chemistry, University of California, Berkeley, CA, 94720*
*£ Now at Department of Environmental Science and Engineering, Gillings School of*
*Global Public Health, University of North Carolina at Chapel Hill*
*† Now at Department of Chemistry, University of North Carolina at Chapel Hill*
*§Deceased April 2017*
May 2018
*Atmospheric Measurement Technology*
*[*]Corresponding authors:* Paul Davidovits, (617)552-3617, davidovi@bc.edu
Charles E. Kolb, (978)663-9500 x 290, kolb@aerodyne.com

**Abstract**

Glass transitions from liquid to semi-solid and solid phase states have important implications for reactivity, growth, and cloud forming (cloud condensation nuclei and ice nucleation) capabilities of secondary organic aerosols (SOA). The small size and relatively low mass concentration of SOA in the atmosphere make it difficult to measure atmospheric SOA glass transitions using conventional methods. To circumvent these difficulties, we have adapted a new technique for measuring glass forming properties of atmospherically relevant organic aerosols. Aerosol particles to be studied are deposited in the form of a thin film onto an interdigitated electrode (IDE) using electrostatic precipitation. Dielectric spectroscopy provides dipole relaxation rates for organic aerosols as a function of temperature (373 to 233K) that are used to calculate the glass transition temperatures for several cooling or heating rates. IDE-enabled broadband dielectric spectroscopy (BDS) was successfully used to measure the kinetically controlled glass transition temperatures of aerosols consisting of glycerol and four other compounds with selected cooling/heating rates. The glass transition results agree well with available literature data for these five compounds. The results indicate that the IDE-BDS method can provide accurate glass transition data for organic aerosols under atmospheric conditions. The BDS data obtained with the IDE-BDS technique can be used to characterize glass transitions for both simulated and ambient organic aerosols and to model their climate effects.

**Keywords**

Broadband Dielectric Spectroscopy          Glass Transition          Organic Aerosols

Interdigitated Electrodes          Thin Films          Aerosol Climate Effects

## 1 Introduction

Aerosol particles have important climate and health effects because they can scatter sunlight, form clouds by acting as cloud condensation nuclei (CCN), alter visibility, and affect human health (Hallquist et al., 2009; Jimenez et al., 2009). Recent studies have confirmed that organic aerosols, which comprise approximately half of the total submicron aerosol mass in the atmosphere, can change from liquid to glassy state at ambient humidity levels and temperatures (Zobrist et al., 2008; Virtanen et al., 2010; Shrestha et al., 2014; Zhang et al., 2015). The effect of temperature may be especially important when aerosol particles are lifted into the free troposphere, where the temperature change can rapidly alter their phase from liquid to glass (Koop et al., 2011). The physical state of the aerosol strongly influences air quality and aerosol climate effects. Evidence suggests that secondary organic aerosols, formed through oxidation of gas phase organic compounds, have much lower vaporization rates than previously assumed, which changes the reactivity of the gas phase species as well as their fate in the atmosphere. The phase state of aerosol particles also influences the diffusion of the gas phase species into the atmosphere, affecting the oxidation extent and multiphase reactions of the particles. For example, Shiraiwa and Seinfeld (2012) used models to predict that when aerosol particles are in certain semi-solid and glassy phase states, the reactive uptake of gas phase species will be kinetically limited. Kuwata and Martin (2012) showed that the phase state of secondary organic aerosols (SOA) affects the uptake of ammonia into the particles. Zhang et al. (2018) provided experimental and modeling evidence that the reactive uptake of isoprene-derived epoxydiols (IEPOX) into acidic sulfate particles is influenced by the phase state, which can contribute to at least a 30% reduction of isoprene-derived SOA in the Southeast U.S.

The phase state of the aerosols also affects their climate properties. For example, in the

glassy state, the water vapor uptake by the SOA is greatly reduced, limiting the ability of
particles to have liquid water condense on them and thus hampering the formation of liquid
cloud droplets (Shiraiwa et al., 2011; Zobrist et al., 2011; Price et al., 2015). However, there is
evidence that glassy SOAs are effective ice nucleation agents. Their ability to nucleate ice
crystals to form Cirrus clouds in the upper troposphere may be particularly important given the
key role of these clouds in global warming (Wilson et al., 2012; Berkemeier et al., 2014). Lack
of adequate data describing these processes contributes to the high uncertainty of atmospheric
aerosol impact on climate change (Shiraiwa et al., 2017).

The importance of the phase state of organic aerosols in the evaluation of their climate

effects has motivated several studies in this field. However, these studies are difficult to perform
and the data obtained so far are limited. Renbaum-Wolff et al. (2013) studied the phase state and
viscosity of the water-soluble part of $\alpha$-pinene SOA at several humidity levels and phase
separation effects. Zhang et al. (2015) characterized the viscosity of $\alpha$-pinene SOA across a wide
range of relative humidity levels. Rothfuss and Petters (2017) studied the viscosities of sucrose
particles with sodium dodecyl sulfate (SDS) particles up to $10^7$ Pa under sub-freezing
temperature regimes. There have also been a few studies exploring the glass transition
temperature of atmospherically relevant organic compounds by using differential scanning
calorimetry (DSC) (Koop et al., 2011; Lienhard et al., 2012; Dette et al., 2014; Dette and Koop,

2015).

Despite past studies, very little information is available on how organic aerosols become

glass as temperature, and the rate of cooling/heating changes. Such information is required to
model the aerosol phase when aerosols are transported from one region of the atmosphere to
another (Murray et al., 2010; Wilson et al., 2012). In an early 2011 study Koop and co-workers
performed experiments that led them to estimate glass transition temperature ($T_g$) values of 268-
290K for a range of surrogate biogenic SOA compounds by utilizing the DSC method. The
results show that oxidation and/or oligomerization reactions leading to higher oxygen to carbon
ratios (O:C) yield higher $T_g$ values. Dette et al. (2014) used the "metastable aerosol by the low
temperature evaporation of solvent" (MARBLES) technique to provide information on the glass-
to liquid transition temperatures of pure organic compounds and organic-inorganic binary
mixtures. Their results show that the glass transition temperatures of these mixtures can be
accurately described by the Gordon-Taylor equation that describes the glass transition of binary
mixtures. However, to evaluate the impact of SOA and its possible phase transitions on climate
and air quality issues, the current techniques need to be improved in order to adapt to the
atmospheric aerosol sampling requirements.

The small particle size and relatively low concentration of SOA in the atmosphere make

it difficult to measure atmospheric SOA glass transitions using conventional methods. First, a
reliable measurement of glass transitions with currently used techniques requires a relatively
large mass, typically milligram levels of the compound, while reasonable field collection
methods yield organic aerosols in the femtogram mass range (Dette et al., 2014; Dette and Koop,
2015). Second, it is difficult to collect suspended aerosols and transfer them to the analysis
apparatus without contaminating the sample with trace water. Trace amounts of water absorbed
by SOA can substantially alter glass transition properties (Bateman et al., 2015; Price et al.,
2015; Rothfuss and Petters, 2017; Shiraiwa et al., 2017). To circumvent these difficulties, we
have adopted a new technique for measuring glass forming properties of atmospherically
relevant organic compounds. The technique combines broadband dielectric spectroscopy (BDS)
utilizing interdigitated electrodes (IDE) (Chen et al., 2012) with organic aerosol sample
deposition using electrostatic precipitation (Liu et al., 2013).

BDS is one of the most widely used techniques for measuring the dynamics and glass

transition of liquid and semi-solids (Richert, 2014). In the usual arrangement, dielectric
spectroscopy instruments consist of two parallel metallic plates with the sample filling the space
between the plates. As was stated, the traditional dielectric method usually requires mass in the
milligram range to perform the measurement (Richert, 2014). Such high mass loading cannot be
reasonably attained with aerosol collected under normal atmospheric conditions. A relatively
new technique, using interdigitated electrodes (IDE), which requires only one surface for
samples and requires mass only in the femtogram range (Chen et al., 2012), is suitable for
atmospheric aerosol phase studies. A thin film is deposited on the IDE first, then the dielectric
spectra are recorded to characterize the glass transition of aerosol particles at variable cooling or
heating rates.

The purpose of this study is to demonstrate the new IDE-BDS analysis technique by

presenting results of the glass transition of SOA surrogates using this technique. In section 2
below, we first describe the experimental setup including aerosol generation, thin film deposition
on the IDE, temperature conditioning chamber, and the BDS measurement system. Then data
analysis, including glass transition determination, is discussed in section 3. Section 4 includes
discussion of the advantages of the IDE-BDS method, as well as caveats associated with its
current implementation.
**2. Experimental Setup**

A schematic diagram of the experimental setup is shown in Figure 1. The setup is

conveniently divided into four parts:1. Aerosol sample generation, 2. Thin film formation via
electrostatic precipitation on the Interdigitated electrodes (IDE) with associated humidity control
3. Temperature conditioning chamber and 4. Broadband Dielectric Spectroscopy measurement
system.
**2.1 Aerosol generation.**
Two types of aerosol generation systems are used in our experiments. The first is a home-
made self-nucleation generation device used for producing liquid organic aerosol samples
including glycerol, 1,2,6-hexanetriol, di-n-butyl phthalate, and dioctyl phthalate. About 0.5 gram
of the glycerol is placed at the bottom of a round flask and the temperature of the flask is heated
to 20°C below the boiling temperature of the organic liquid. A condenser is connected to the top
of the flask to cool the temperature of that region. A flow of 2 liters per minute (Lpm) of dry air
passes through the condenser and brings the aerosol particles to the region where the aerosols are
precipitated onto the IDE.
The second method utilizes a commercial unit (TSI, 3076) to generate atomized citric
acid aerosols. About 0.5 gram of citric acid is dissolved in 100 mL of high purity water to form
the atomizing solution. About 30 psi pressure of dry air is applied on one end of the atomizer to
generate a constant 3 Lpm aerosol-containing flow to the second part of the system, which is the
thin film generation system that will be described below.
**2.2 Interdigitated electrode (IDE) and thin film formation.**
An IDE (NIB003744, MS-01/60, NETZSCH Instrument North America) is used in this
study as a substrate for measuring the dielectric constants of organic materials. The IDE consists
of two thin electrodes that are interdigitated together like entwined finger tips, as shown in
Figure 2. Each interdigitated pair serves as a small capacitor for dielectric analysis. The thin
electrodes are made from platinum (Pt) and are arrayed on a quartz substrate. The electrodes
utilized in this study are spaced 1 μm apart and are able to withstand temperatures up to 200 °C.
The combination of multiple interdigitated pairs of electrodes greatly enhances the sensitivity of
the technique compared to a single pair of electrodes that would have been used in the
conventional technique.
An electrostatic deposition method is used to deposit organic films on the IDE (Liu et al.,
2013). The electrostatic precipitator has one inlet and one outlet. A stream of aerosolized
oxygenated organic liquid droplets to be studied is passed through an inlet with a high voltage
corona discharger (-5000V) so that all the droplets are negatively charged to varying degrees. The
flow is directed above the substrate held at +3000 V within the precipitator. Due to opposite
charges, the charged particles are electrostatically deposited onto the substrate, gradually merging
together to form thin films. The remaining flow is then withdrawn from the precipitator and flows
through a HEPA filter connected to a pump. The flow rate through the precipitator is maintained
between 1.7-1.9 Lpm. Depending on the amount of aerosol material being deposited onto the
surface, the deposition can remain either remain as discrete aerosol droplets, or at higher droplet
depositions, can form a uniform thin film, as shown in Figure 2. For this study thin films are
formed. The volume concentration of the aerosol particles at the inlet of the precipitator was
$4.5 \times 10^{11}$ nm$^3$ cm$^{-3}$. After collecting for 5 hours, the film thickness is estimated to be 1-2 μm on a
1 mm × 8 mm substrate based on the difference of volume concentration between the inlet and the
outlets of the precipitator, the flow rate, the collection time, and assuming 50% collection
efficiency.
**2.3 Temperature conditioning chamber.**
The IDE substrate coated with organic material is then transported to the temperature
conditioning chamber using tweezers. The temperature conditioning chamber consists of a
stainless steel cap and a heating/cooling surface using either a liquid nitrogen cooler or a heating
furnace. The sample temperature can be controlled from ~-150°C to +200°C. Details of the
chamber are shown in Figure 1. The cooling/heating rate can be varied from 1 K/min to 25
K/min. The chamber is flushed with dry nitrogen gas to reduce the relative humidity (RH) prior
to temperature conditioning. A K type thermocouple is located on top of a reference cell inside
the conditioning chamber, to monitor sample temperature. The typical cooling cycle starts
around 20 °C and ends at about -140 °C, while the heating cycle starts at -140 °C and ends at 30
°C. The cooling and heating cycle are adjusted to the desired cooling/heating rates between 2
K/min and 10 K/min.
**2.4 Broadband Dielectric Spectroscopy (BDS) measurement system.**
The BDS instrument used in this study is manufactured by NETZSCH Inc. (DEA 288
model). A periodic signal from the instrument is applied to the IDE electrodes. The frequency of
the signal ranges from $10^{-3}$ Hz to 1 MHz. The data acquisition part of the instrument then
measures the impedance, $Z_{sample}$, of the sample as a function of the applied frequency. The
impedance measurement yields the capacitance of the sample on top of the IDE. By measuring
the impedance of the uncoated and organic-coated IDE, three IDE capacitances can be obtained,
i.e., when the IDE is uncoated, $Z_{empty}$, when it is coated with organic compounds, $Z_{coated}$, and the
geometric capacitance of the IDE without any substrate, $Z_{geo}$. Approximate values of the real and
imaginary part of the sample permittivity, $\varepsilon_{sample}$, can be obtained using Eq. (1) (Chen et al.,

2012)

$$\varepsilon_{sample} = 1 + \frac{Z_{loaded} - Z_{empty}}{Z_{geo}} \qquad (1)$$

For demonstration and data comparison purposes, we have used glycerol, 1,2,6-
hexanetriol, di-n-butyl phthalate, and dioctyl phthalate (99%, Sigma Aldrich, St. Louis, MO,
USA) as the test compounds for homogeneous nucleation and citric acid (99%, Sigma Aldrich,
St. Louis, MO, USA) as the surrogate organic aerosol generated by atomizing solutions. For
atomizing solutions, the surrogate compound is mixed with high purity water. All reagents were
used as provided without further purification.
**3. Data Analysis**
**3.1 Calculating Relaxation Time $\tau$**
The thin film on the IDE is usually cooled at a selected cooling rate and then heated back to 30°C.
After a cooling-heating cycle, the dielectric constant at each temperature measured, ε (ω), is
recorded by instrument. The relaxation time, τ, can be obtained by curve fitting the Havriliak-
Negami equation of the real and imaginary parts ($\varepsilon'(\omega)$ and $\varepsilon''(\omega)$, respectively) with the
frequency $\omega$, as shown in Figure 3 (Chen et al., 2012). The detailed equation for $\varepsilon'(\omega)$ and $\varepsilon''(\omega)$
is shown in Eqns. (S1) and (2). At each temperature, the dielectric spectra often show peaks at
specific frequencies, designated as dielectric relaxation peaks. Different peaks give different τ
values after fitting Eq. (2) with the data points.

$\varepsilon''(\omega) = \Delta\varepsilon(1 + 2(\omega\tau)^\alpha \cos\left(\frac{\pi\alpha}{2}\right) + (\omega\tau)^{2\alpha})^{-\beta/2}\sin(\beta\varphi)$           (2)

with $\Delta\varepsilon = \varepsilon_s - \varepsilon_\infty$, $\varphi = \arctan\left(\frac{(\omega\tau)^\alpha \sin\left(\frac{\pi\alpha}{2}\right)}{1+(\omega\tau)^\alpha \cos\left(\frac{\pi\alpha}{2}\right)}\right)$

where $\varepsilon_s$ is the permittivity at lower frequency, $\varepsilon_\infty$ is the permittivity at the high frequency limit,
$\alpha$, $\beta$ are fitting parameters, and $\tau$ is the characteristic relaxation time of the medium
(Adrjanowicz et al., 2009; Chen et al., 2012).

Log $\tau$ is then plotted as a function of the inverse of the temperature to further examine

how relaxation time changes as a function of temperature. The error bar represents twice the
standard deviation of the fitting result. The resulting curve can be used to calculate the glass
transition temperature of the compound, as described in section 3.2.
**3.2 Glass Transition Determination**

The glass transition temperature is defined as the temperature where a compound changes

from liquid to glass. Several methods have been used to indirectly determine the glass transition
temperatures. A common way to calculate the glass transition temperature using BDS is to measure
relevant parameters and to calculate the dielectric relaxation $\tau$ described in section 3.1 at several
equilibrium temperatures T, and then plot $\log \tau$ as a function of 1000/T. The data points are fitted
using the Vogel-Fulcher-Tammann (VFT) formula (Vogel, 1921; Fulcher, 1925; Tammann and
Hesse, 1926). The glass transition is customarily defined as the temperature where $\tau = 100$ s in the
fitted curve (Chen et al., 2012; Richert, 2014). The result usually agrees with the DSC
measurement within a few degrees (Richert, 2014). However the method is limited, because not
all compounds become glass as $\tau = 100$ s (Saiter et al., 2007; Bahous et al., 2014). Furthermore, this
method does not take into account kinetic effects on glass transition, specifically the effect of
cooling and heating rates (Elmatad et al., 2009, 2010; Keys et al., 2013; Limmer and Chandler,
2014; Hudson and Mandadapu, 2018), as glass transition temperature changes with cooling and
heating rates.

The method used in our studies is based on dynamical facilitation theory (Elmatad et al.,

2009; Chandler and Garrahan, 2010; Keys et al., 2011; Keys et al., 2013; Hudson, 2015, Hudson
and Mandadapu, 2018), which also takes into account the effect of cooling rate on the glass
transition.  According to this theory, as a compound is cooled and transitions from a liquid to a
supercooled liquid it exhibits super-Arrhenius behavior given by the following equation:

$\log \tau/\tau_0 = J^2(1/T - 1/T_o)^2$                          (3)

where J is an energy scale intrinsic to each material related to the rate of motion of

individual molecules, $T_o$ is termed the "onset temperature" and refers to the temperature at which
a liquid showing Arrhenius relaxation becomes a supercooled liquid showing super-Arrhenius
relaxation, and $\tau_0$ is a temperature-independent reference time scale of the order of the time taken
for molecules to locally rearrange (Keys et al., 2011).

As temperature further decreases, the supercooled liquid becomes glass-like, exhibiting

Arrhenius behavior. The temperature where the supercooled liquid changes to glass is the glass
transition temperature of the compound at the specific cooling rate studied. As the sample is
continuously cooled at a specific cooling rate, the dielectric relaxation peaks can be generated as
a function of the sample temperature. The experimentally obtained data are plotted in the form
$\log \tau$ vs 1/T. The data obtained at the higher temperature range are fitted to the super-Arrhenius
function and the data obtained at the lower temperature range are fitted to the Arrhenius function.
Figure 4 shows a typical relationship between the dielectric relaxation timescale and the
temperature, as a compound is cooled down or warmed up between the liquid state and glassy
state. As illustrated in Figure 4, the kinetically controlled glass transition temperature (or the true
glass transition temperature) is the temperature at the intersection of the two functions. The
traditional method determines the glass transition temperature as shown in dashed lines where $\tau$
=100 s. Depending on the compound, the true glass transition temperature may not be the same
as the glass transition temperature determined by using $\tau$ =100 s, as shown in section 4 below.
The uncertainty of the glass transition temperature is estimated based on varying the fitting
parameters of the super-Arrhenius curve and Arrhenius line within a one-sigma range.

For glycerol, measurements of glass transitions were performed at three cooling rates: 2

K/min, 5 K/min, and 10 K/min. At each cooling rate the compound is cooled from approximately
300 K to 125 K, while the dielectric peaks are measured simultaneously as a function of
temperature. The organic film is thin enough so that its temperature reaches equilibrium with the
cooling/heating medium, reducing the errors caused by heat transfer within the sample itself.
Such measurements are difficult to perform with conventional techniques due to slow heat
transfer in large mass samples, which often leads to inaccurate results. The effect of cooling rates
on glass transition measurements will be discussed in the following section.
**4 Results and Discussion**
**4.1 Glass transition temperature of selected organic compounds**

Aerosols are generated by two methods in this study, and in each method, we measured

the glass transition of a compound that has been studied in the literature. Glycerol, 1,2,6-
hexanetriol, di-n-butyl phthalate, dioctyl phthalate, and citric acid particles were generated
through the homogeneous nucleation method or the atomizer method, respectively. The
measured dielectric spectra of each compound show distinct relationships of their dielectric
relaxation timescales with a 5 K/min cooling rate, as shown in Figure 5. This is a confirmation of
the expected behavior.

By calculating the corresponding temperature when the super-Arrhenius curve intersects

with the Arrhenius curve, the glass transition of each compound can be derived. As is shown in
Table 1, the kinetically controlled glass transition data agree well with previously measured
literature values of glycerol (Zondervan et al., 2007; Chen et al., 2012; Amann-Winkel et al.,
2013), 1,2,6-hexanetriol (Dorfmüller et al., 1979; Böhmer et al., 1993; Nakanishi and Nozaki,
2010), di-n-butyl phthalate (Dufour et al., 1994), dioctyl phthalate (Beirnes Kimberley et al.,
1986), and measured values are mostly within 4% of the cited literature value, except for one
study by Dorfmüller et al. (1979). The comparison of the dielectric relaxation timescale of
glycerol measured in this study with literature values is shown in Figure S1, for several different
temperatures. The results show that our measurements match the previously published results in
the super-Arrhenius region when the compound is in equilibrium at a given temperature, with
almost identical values. As the temperature continues to drop, glycerol and other compounds we
tested fall out of equilibrium and become glass, which exhibits Arrhenius behavior. The
transition from super-Arrhenius to Arrhenius behavior in this study provides the kinetically
controlled glass transition as the compounds change from liquid to glassy state.
For citric acid, there is no dielectric measurement available. The kinetically controlled
dielectric spectra fitted using Eq. (S2) are shown in Figure 6. From the fitting results in Figure 6,
the dielectric relaxation timescales $\tau$ are derived as a function of temperature, as shown in Figure
7. The kinetically controlled glass transition temperature derived from Figure 7 agrees
reasonably well (within 10% error) with four literature results (Lu and Zografi, 1997; Bodsworth
et al., 2010; Dette et al., 2014; Lienhard et al., 2014). The fifth set of literature data (Murray,
2008) is based on extrapolation of a fit to experimental data and is about 20K lower than other
values reported by the literature. The differences can be explained by the following reasons: (1)
To date there have been no measurements of the dielectric spectra for citric acid to our
knowledge. The citric acid references in Table 1 are based on DSC measurements, which may
not be in exact agreement with the dielectric measurement. Angell (2012) pointed out that there
are at least three different definitions of the glass transition and the Tg determined by each can
be 50 K different from each other. DSC uses heat capacity changes to measure the glass
transition while dielectric relaxation uses molecular movements to define the glass transition.
Due to the difference in measurement parameters and the definition of the glass transition, these
two measurements can provide different Tg values of the same compound by up to 10 K
(Shinyashiki et al., 2008). (2) This study focuses on the kinetically controlled glass transition
temperature at a given cooling rate, i.e., the transition between the Arrhenius and super-
Arrhenius relaxation regimes to determine the glass transition temperature. If the more
conventional method, i.e. fitting the super-Arrhenius curve to obtain the glass transition
temperature when $\tau=100$ s, is applied to the data, as shown in Figure 7, the obtained glass
transition temperature would be $281 \pm 3$ K, which is within 1% difference compared with the
two nearest literature values. However, as previous publications have pointed out (Keys et al.,
2013; Bahous et al., 2014; Limmer and Chandler, 2014; Hudson, 2018), using the transition from
super-Arrhenius to Arrhenius region is likely to reflect the true glass transition when taking
kinetic factors, such as cooling/heating rates, into consideration. (3) Moreover, glass transition
data of citric acid are rather limited and there are differences between each study. For instance,
from the literature data, the differences between three reported glass transition temperatures are
up to 10%, which is about the same difference obtained by comparing our data to the other two
nearest literature results. (4) The heating and cooling rates may also contribute to the difference
of the $T_g$ of citric acid between the literature and this study, as this study uses a lower cooling
rate than the ones reported by the literature. Moynihan et al. (1974) reported that a change of 2
K/min to 10 K/min cooling rate could alter the glass transition temperature of borosilicate by 15
K. It is possible that citric acid is a less fragile liquid, similar to borosilicate, i.e., the glass
transition temperature depends strongly on cooling or heating rates. Therefore, the difference
herein is likely due to including kinetic considerations such as heating/cooling rates in the
measurement of the glass transition. We report the citric acid glass transition temperature as 307
$\pm 5$ K at 5 K/min warming rate, which is likely a more accurate way of reflecting the glass
transition, as the kinetic process is considered, as shown in Figure 7.
Even though the issues listed above are likely to be the primary reasons leading to ~10%
difference of the glass transition temperature of citric acid between our results and the literature,
the following factors may play a role: (1) Atomizing the citric acid solution and re-depositing
citric acid particles via electrostatic precipitation could introduce impurities during the
atomization process that may affect the glass transition temperatures; (2) The glass transition of
citric acid was measured during a warming cycle. A fast non-recordable cooling cycle at
20K/min was performed prior to warming in order to inhibit the citric acid crystal formation. The
hysteresis effect will lead to an increase of the glass transition temperature from the warming
cycle compared with data obtained from the cooling cycle (Wang et al., 2011); (3) Lu and
Zografi (1997) have shown that different ways of preparing the citric acid can lead to differences
in glass transition measurements. The thicknesses of the thin films are equal to or less than one
micrometer, leading to confinement effects and differences in glass transition temperature
measured from bulk compounds (Park and McKenna, 2000). The results show that the thin film
IDE-BDS method can accurately measure the glass transition temperatures of various organic
compounds that are comparable to the composition of organic aerosols.
**4.2 The influence of cooling rates on glass transition temperatures**
One advantage of this study is the introduction of cooling and heating rates as variables for
glass transition temperature measurement for organic compounds. For BDS studies, the glass
transition temperature of a compound is often deduced by measuring the sample at a few
isothermal temperatures and fitting the curve of temperature and relaxation time in order to identify
the temperature when relaxation time corresponds to 100 s. This traditional approach makes it
difficult to directly compare the result with glass transition temperatures deduced from DSC
studies with variable cooling rates. One of the advantages of our technique is that variable cooling
rate measurements are performed on the thin film and the cooling rate dependent glass transition
temperature of target species is determined.

The influence of cooling rate on glass transition temperature was carefully examined by

repeating the glycerol experiment at two additional cooling rates. The resulting super-Arrhenius

curve for each cooling rate is plotted against the Arrhenius lines of that cooling rate, as shown in

Figure 8. As a compound remains in the supercooled liquid stage, the relaxation time is short

enough that it is constantly in equilibrium with external perturbations, leading the super-Arrhenius

region totally reversible so it should behave the same for all cooling rates. The super Arrhenius

part of the data from all three different cooling rates all collapse into one single trend, indicating

the data collected agree well with the theory. The super-Arrhenius curve in Equation (3) from

Elmatad et al. (2009) is also plotted as the black dashed line and it also agrees well with our

experimental results. As the cooling continues, the relaxation time gets longer until it cannot keep

up with the external temperature change, leading to the compound falling out equilibrium and

forming a glass. Therefore, a faster cooling rate often leads to a quick falling out of equilibrium

and a higher glass transition temperature for the compounds studied, as demonstrated in Figure 8.

Based on the intercept of the super-Arrhenius curve and the Arrhenius line, the glass transition

temperatures for 5 K/min and 10 K/min cooling are determined to be $192 \pm 2$ K and $194 \pm 2$ K.

For 2 K/min cooling, because an Arrhenius line does not appear within our range of measurement,

the glass transition temperature will likely be lower than 189 K. This study reports an increase of

5 K or more in the glass transition of glycerol as the cooling rate changes from 2 K/min to 10

K/min. The reported increase in glass transition temperature during these cooling rates agrees with

the behavior of sorbitol and fructose in another study performed by Simatos et al. (1996), which

also measured the dependence of glass transition temperature on cooling rate with small organic

molecules.

Our results agree reasonably well with other studies for the glass transition temperatures
of the five compounds chosen.  However, there are a couple possible caveats for this study. One
is the influence of humidity during the cooling process. Even though dry nitrogen is used to flush
through the cooling chamber to remove any extra water vapor present prior to cooling, there is still
the possibility that the chamber wall surface can degas and release water vapor to the system during
the cooling process. The effect of water vapor on the testing materials is likely to be small, but
should be considered when the organic compound tested can readily absorb water at low RH
conditions. The other potential caveat is the influence of non-equilibrium heat transfer within the
thin film on the measurement of the glass transition temperature. Because the sample is being
cooled from underneath, the heat transfer between the upper and lower boundaries of the film can
lead to uncertainties in measuring the glass transition temperature. The organic thin films made
during the experiments are within the micrometer range, therefore the thermal gradient across the
film is small enough to be likely insignificant compared with other systematic errors. Moreover,
theoretical models to predict the glass transition of compounds with variable cooling rates are
needed in further studies to verify and explain these experimental measurements.
**5 Summary**
In this work, we have demonstrated a novel method using interdigitated electrodes,
broadband dielectric spectroscopy, and electrostatic precipitation together as an efficient and
powerful approach studying the phase and glass transitions of organic particles under various
cooling rates. The method is particularly suitable for studying the glass transition of submicron
organic particles whose mass loading is generally too small for other kinds of glass transition
measurement techniques. The results from this technique agree well with published studies using
other methods. Future publications will report glass transition measurements for simulated SOA
mixtures as well as laboratory produced SOA particles.

The dielectric relaxation peaks of glycerol and four other compounds were recorded, and

the logarithm of characteristic relaxation time were calculated and plotted as a function of inverse
temperature. The transition between the super-Arrhenius and Arrhenius curves were used to
determine the temperature where the super-cooled liquid fell out of equilibrium to become glass,
which is defined as the true experimental glass transition temperature. Furthermore, cooling rates
are demonstrated to have an effect on the glass transition temperature. By changing the cooling
rate from 2 K/min to 10 K/min, the glass transition temperature increases by at least 5 K for
glycerol.
**Acknowledgments** We acknowledge James Brogan, Yatish Parmar, Leonid Nichman, Professor
Ranko Richert, Lindsay Renbaum-Wolff, Wade Robinson, Paul Kebabian, Professor Jason D.
Surratt, and Professor Andrew Ault for useful discussions and assistance with the experiments.
**Funding** This material is based upon work supported by the National Science Foundation
Environmental Chemistry Program in the Division of Chemistry under Grant No. 1506768, No.
1507673, and No. 1507642.
**Competing financial interests:** The authors declare no competing financial interests.

**Table 1.** Glass transition temperatures for selected organic species measured by broadband

dielectric spectroscopy with a thin-film interdigitated electrode array

| Compound | Chemical Formula | $T_g$ (K)-Measured | $T_g$ (K)-Literature |
|---|---|---|---|
| Glycerol | $C_3H_8O_3$ | <189 K (2 K/min) <br> 192 ± 2 K (5 K/min) <br> 194 ± 2 K (10 K/min) | 190 K (Zondervan et al., 2007) <br> 191 K (Chen et al., 2012) <br> 196 K (Amann-Winkel et al., 2013) <br> 191.7 ± 0.9 K (Lienhard et al., 2012) |
| 1,2,6-Hexanetriol | $C_6H_{14}O_3$ | 192 ± 2 K (5 K/min) | 200 ± 2 K (Nakanishi and Nozaki, 2010) <br> 206.4 ± 0.5 K (Dorfmüller et al., 1979) <br> 202 K (Böhmer et al., 1993) |
| Di-n-butyl Phthalate | $C_{16}H_{22}O_4$ | 180 ± 2 K (5 K/min) | 174 K (Dufour et al., 1994) |
| Dioctyl Phthalate | $C_{24}H_{38}O_4$ | 194 ± 2 K (5 K/min) | 190 K (Beirnes Kimberley et al., 1986) |
| Citric Acid | $C_6H_8O_7$ | 307 ± 5 K (5 K/min) | 281 ± 5 K (Bodsworth et al., 2010)[*] <br> 285 ± 0.2 K (Lu and Zografi, 1997) <br> 281.9 ± 0.9 K (Lienhard et al., 2012) <br> 283-286 K (Dette et al., 2014) <br> 260 ± 10 K (Murray, 2008)[**] |

[*] The data was based on modeling result.  [**]The data was based on extrapolation of a fit to the

data.

**List of Figures**

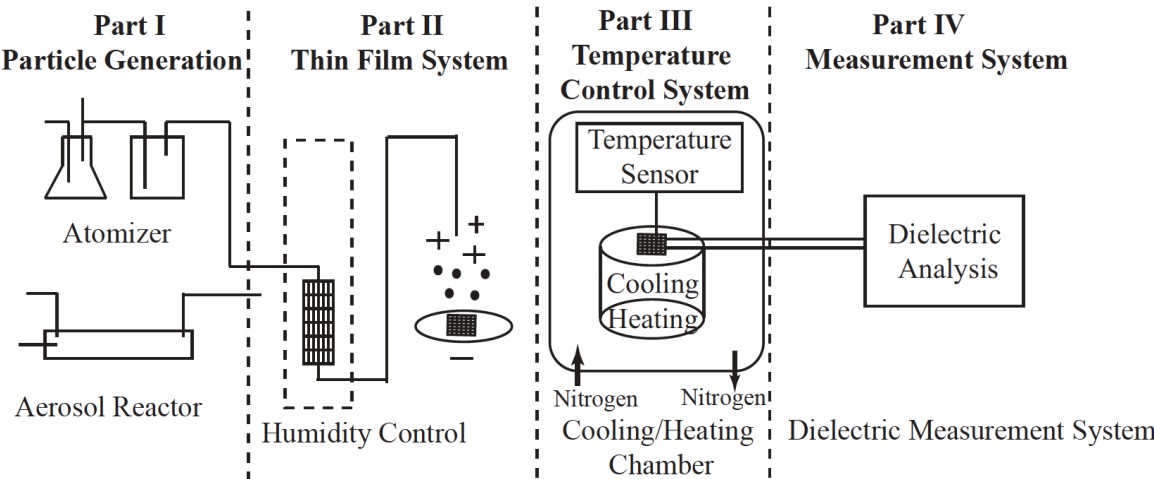

**Figure 1.** A schematic diagram of the experimental setup and procedure. The experimental approach consists of four parts: aerosol generation, thin film deposition using an electrostatic precipitator, the temperature control system, and the broadband dielectric spectroscopy measurement system.

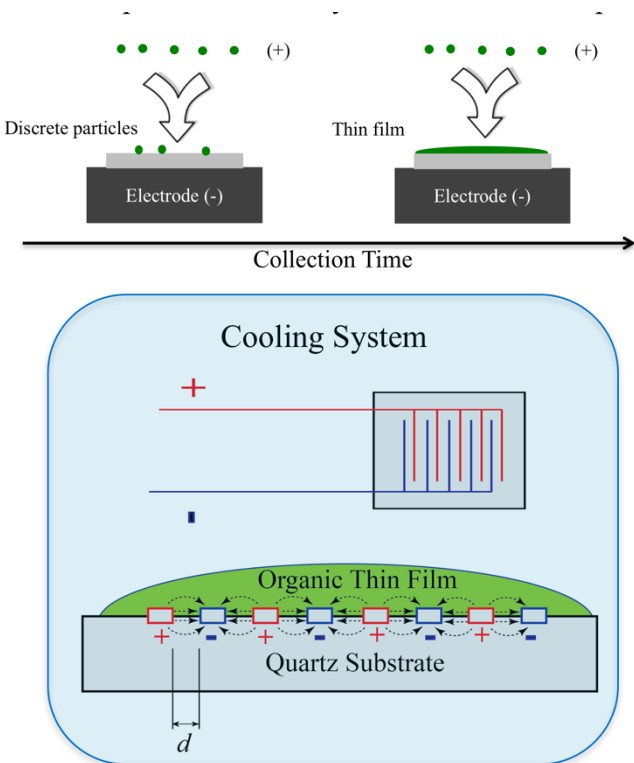

**Figure 2.** A schematic diagram of the key experimental setup. The upper panel shows the

formation of organic thin films though electrostatic deposition. The lower panel shows the

working principle of broadband dielectric spectroscopy (BDS) with an interdigitated electrode

sensor. A compound is placed on an array of interdigitated electrodes with a periodic voltage,

and the compound's impedance is recorded.

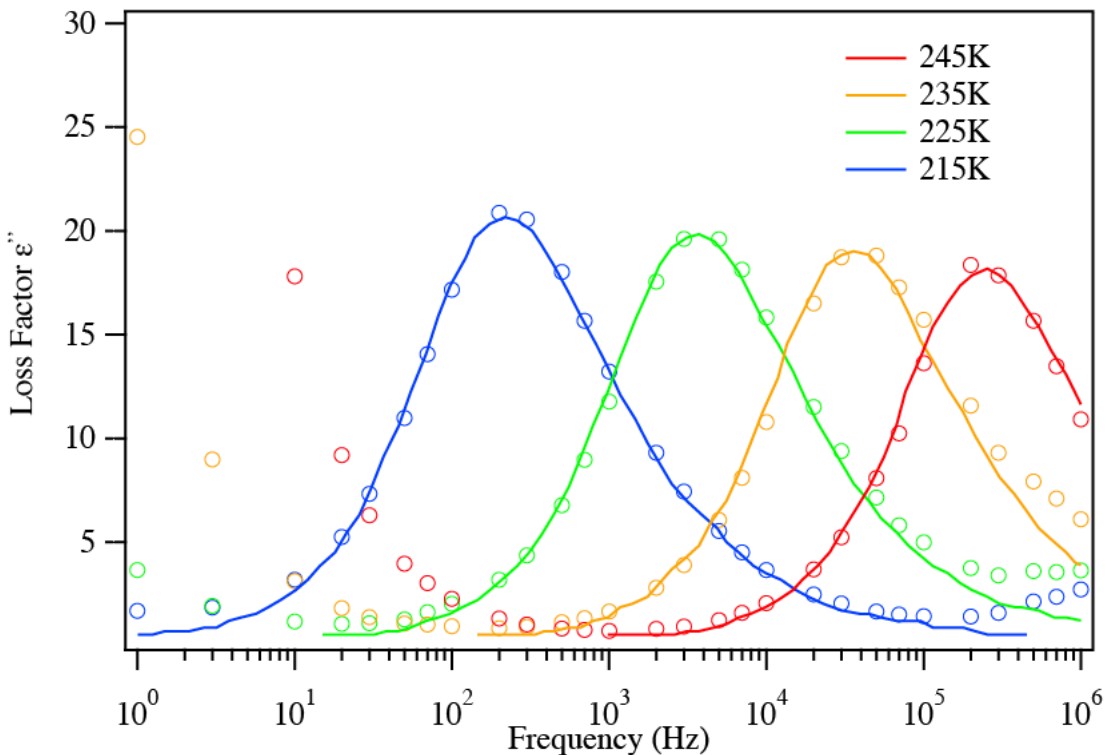

**Figure 3.** The dielectric relaxation spectrum of glycerol at different temperatures. The open circles are measured experimental data and the solid lines are literature data from Chen et al. (2012). As temperature decreases, the dielectric peaks shift towards lower frequencies, indicating that the relaxation timescale increases.

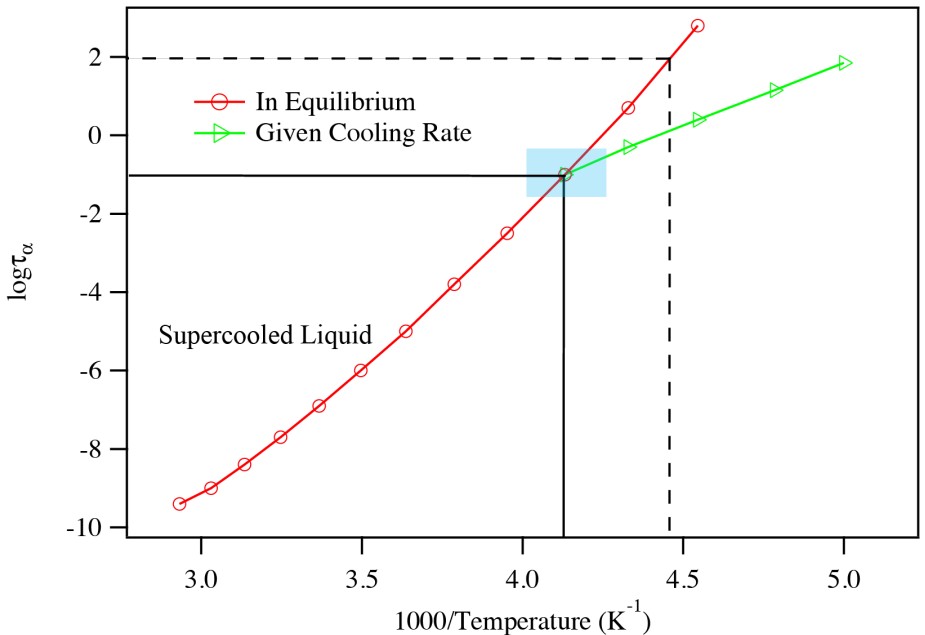

**Figure 4.** An illustrated plot of the relationship between dielectric relaxation time scale and temperature. The logarithm of the relaxation is plotted against inverse T, i.e., log τ vs 1000/T. By linking the data points together, one can plot the super-Arrhenius curve (red) and the Arrhenius line (green). Using a consistent cooling rate, the intersection of the two regions identifies the compound's glass transition temperature, as indicated by the shaded blue region. The intersection of the two black lines represents the glass transition point. The intersection of the two black dashed lines shows the glass transition temperature determined using the traditional method of identifying the temperature when $\tau$=100 s.

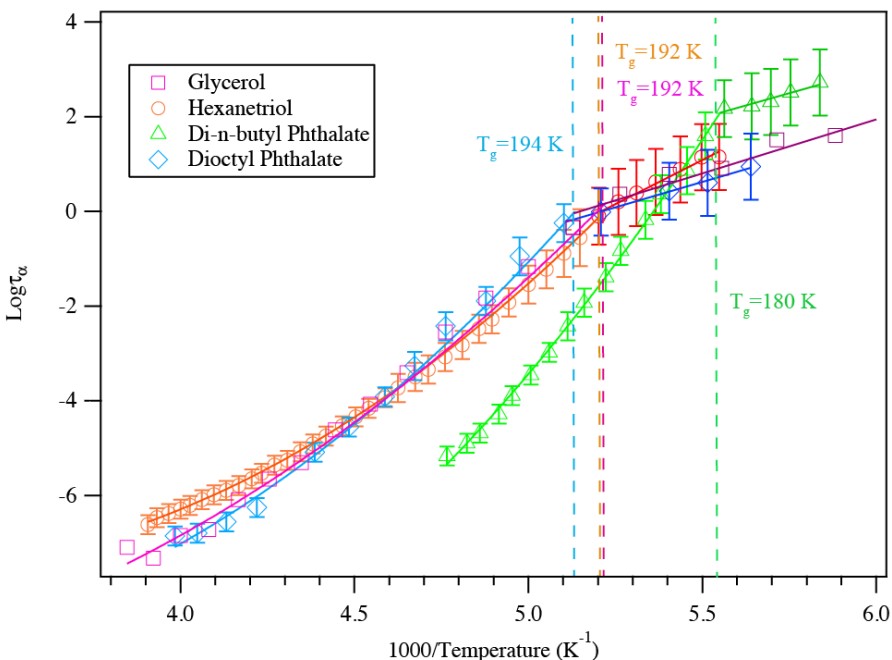

**Figure 5.** A plot of superimposed data points and curves constructed for glycerol, 1,2,6-hexanetriol, di-n-butyl phthalate, and dioctyl phthalate cooled at 5K/min. The solid color lines represent the fitted curves for the super-Arrhenius and Arrhenius region. The intersection between the two lines indicates the kinetically controlled glass transition region for each compound. The glass transition at a 5K/min cooling rate for each compound is shown in the plot.

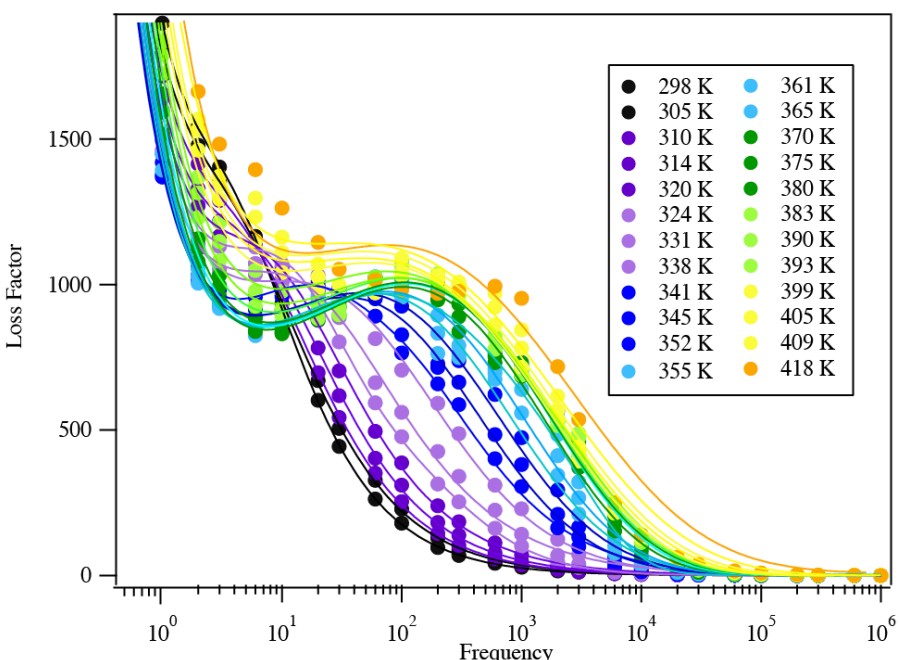

**Figure 6**. The dielectric relaxation spectrum of citric acid at different temperatures. The solid circles are measured experimental data and the solid lines are fitted curves parameterized from Eq. (S2) and Adrjanowicz et al. (2009).

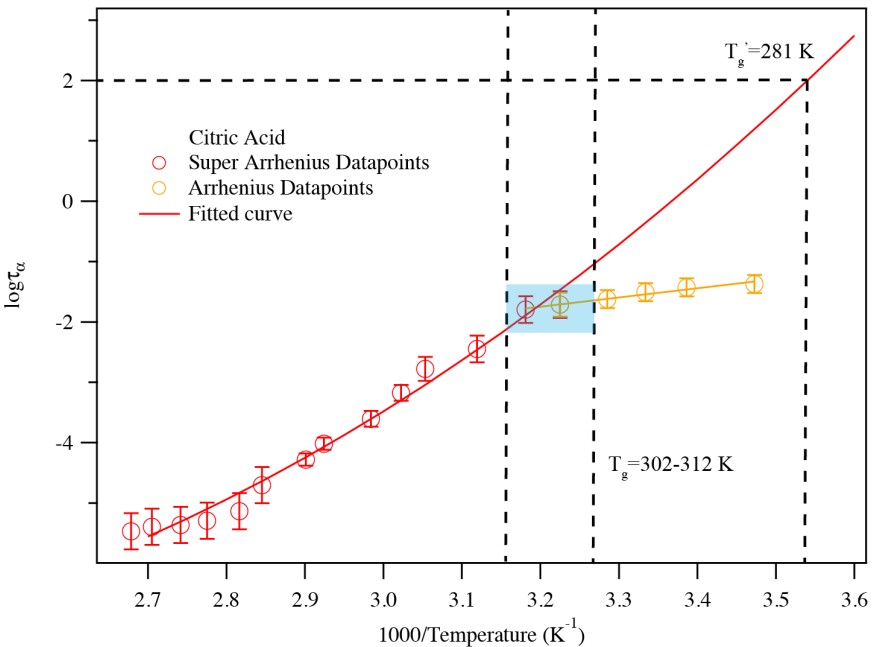

**Figure 7.** A plot of superimposed data-points and curves constructed for citric acid warmed at 5K/min. The solid color lines represent the fitted curves for the super-Arrhenius and Arrhenius region. The blue shaded area shows the glass transition region. The two vertical black lines associated with the blue shaded area indicate the corresponding temperature range where the super-Arrhenius curve intersects with the Arrhenius line. The traditional glass transition temperature, i.e., the temperature when $\tau$=100 s, is also marked.

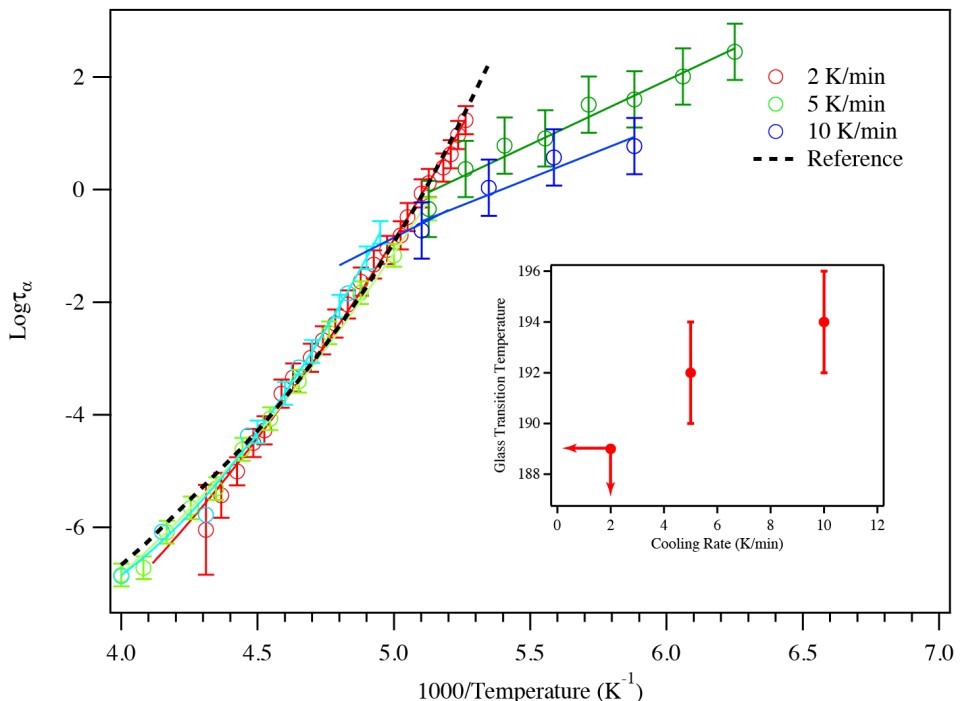

**Figure 8.** A plot of superimposed data-points and curves constructed for glycerol cooled at 2K/min, 5K/min, and 10K/min. The black dashed line represents literature data of the super-Arrhenius region from Elmatad et al. The open circles are experimental data and the solid lines are fitting results. The inset shows the glass transition temperature of glycerol as a function of cooling rates. At 2 K/min cooling, the glass transition temperature has a higher bound of 189 K.

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
