# Peer review of "SUPPORTING INFORMATION"

_Atmospheric Measurement Techniques, 2017_

## Referee Comment (RC1) · Anonymous Referee #1 · 26 Feb 2018

Zhang et al. present a new experimental setup measuring glass transition temperature (Tg) using dielectric spectroscopy of thin films. Aerosol particles are deposited on an interdigitated electrode device using an electrostatic precipitator, eventually forming a thin film. The authors propose a new data analysis approach using the broadband dielectric spectroscopy data to determine Tg. The advantage of the technique compared to more traditional calorimetric techniques is that it needs considerably less material, hence it may open the possibility to measure ambient aerosol. Clearly, that makes the paper important and well suited to be published in AMT. The authors use two surrogates for secondary organic aerosol (SOA) to demonstrate that their technique yields glass transition temperatures comparable to reference methods. Both are pure

compounds (glycerol and citric acid), no simple mixture nor an aqueous solution was investigated.

However, as one of the two tests (citric acid) show quite substantial deviation in measured Tg (307 K) compared to what is measured with calorimetric techniques (281-285 K) -when the data are analyzed using the new technique - there is little evidence (namely, only the glycerol data) that the technique will actually work for ambient or laboratory SOA particles. As there are ample data available for pure compound SOA surrogates, aqueous SOA surrogates, as well as for simple mixtures [e.g. Lienhard et al., 2012; Dette et al., 2015; Dette and Koop, 2015], I think it is essential to perform measurements on more pure component surrogates, at least one mixture and – if possible – also an an aqueous surrogate to characterize the experimental setup. As it is now, the paper does not allow the reader to judge whether the technique is actually feasible or needs further improvements before being applied to samples of unknown Tg.

Detailed comments:

1 Introduction:

I suggest add a paragraph explaining that one important property needed to understand kinetic limitations in gaseous uptake or loss of compounds in atmospheric aerosol particles is the diffusivity of this compound in the condensed phase. Measuring Tg of ambient SOA will indicate that below/at this temperature kinetic limitations will occur, but it does not give immediate quantitative insight.

Technical comment: Page 3, line 60: "nucleate" seems not the right wording here, better use "become activated" or similar.

2 Experimental Setup:

One information missing is how much aerosol mass is needed for producing the film on the device. At least the area of the device and the approximate film thickness should

be given. Even better, if the authors can provide the mass concentration in the aerosol flow prior to the precipitator and the time needed to accumulate for forming the film on the device.

3 Data analysis:

I do not see the advantage of not adding the information of the SI into the main text. Please. Incorporate it including the raw data of Fig. S1 into section 3.1.

I also find section 3.2 difficult to read as a non-expert. How does the discussion here connect to the issues of determining "fragility" [e.g. Angell, 2002]? I am not convinced that the glass community agrees that the intercept shown in Fig. 4 is the "true" glass transition.

Fig. 4: The caption needs to say which material is shown. Or, is this just a sketch to show the idea? Instead of "given cooling rate" in the legend, the actual cooling rate should be provided. They are no error bars shown for the data points: Are these actually the measurements, or some points arbitrarily taken from the fits? You need to provide more detail here.

4 Results and discussion:

First, there is more data available in the literature to compare your data with: for glycerol see for example Lienhard et al. (2012). Citric acid has been measured by Lienhard et al. [2012] as well and by Dette et al. [2014].

Second, in Fig. 5 you determined Tg of citric acid as 305-315 K, whereas in Table 1 you write 307+-5 K. Please check.

Obviously, Fig. 5 shows a very significant difference when comparing the "classical" determination (tau = 100s) for Tg (281 K) and your new method (305-315 K). As the former agrees with the calorimetric measurements of numerous experiments in the literature, whereas the latter show significant deviation, you definitely need more evidence than provided to support the new method. At present, the reader will conclude

that the new method is not reliable.

Concerning the influence of cooling rates, I recommend to the authors to make use of the detailed study of Simatos et al. [1996], do an additional experiment using sorbitol or fructose with the setup, and compare the results with those of Simatos et al. [1996].

References:

Angell, Chem. Rev., 102, 2627-2650 (2002).

Dette et al., J. Phys. Chem. A 118, 7024-7033 (2014).

Dette and Koop, J. Phys. Chem. A 199, 4552-4561 (2015).

Lienhard et al., J. Chem. Phys. 136, 074515 (2012).

Simatos et al., J. Thermal Anal. 47, 1419-1436 (1996).

---

## Referee Comment (RC2) · Anonymous Referee #2 · 28 Mar 2018

This study developed a method to measure the kinetically-controlled glass transition temperatures using the broadband dielectric spectroscopy. Aerosol particles are deposited in the form of a thin film on an electrode using electrostatic precipitation. Glass transition temperatures of glycerol and citric acid were measured, agreeing reasonably well with available literature data. By considering the intersect of super Arrhenius and Arrhenius lines, the new method appears to provide more reliable glass transition temperatures. The effect of cooling rates on glass transition temperatures were also investigated. I found that the measurements were conducted in a highly elegant way. The manuscript is clearly written and easy to follow. In fact, I enjoyed reading this manuscript very much.

[Figure]

My only major concern is that the method was validated with only two compounds. The authors plan to apply this method to measure glass transition temperatures of highly complex SOA mixtures in follow-up studies (L342). In this sense, it would be essential to test a few more compounds. There are indeed a number of organic compounds with known Tg, and I wonder why the authors would not have conducted some more measurements to consolidate this new method. Is this method too time-consuming or not so easy to be applied with more compounds?

Minor comment: - L61: I suggest citing also Zobrist et al., PCCP, 13, 3514, 2011.

- L72: Rothfuss and Petters, PCCP, 2017 measured the viscosity of sucrose but not various types of SOA (or did they?).

- L282: "the" should be replaced to "that".

- SI contains useful information (especially Figure S1 is interesting) and I suggest moving them into the main manuscript or appendix.

---

## Author Comment (AC1) · 4 May 2018

Please see the attached response to reviewers and the revised manuscript. Thank you!

Please also note the supplement to this comment:
https://www.atmos-meas-tech-discuss.net/amt-2017-425/amt-2017-425-AC1-supplement.pdf

———————————————

---

## Author Comment (AC2) · 4 May 2018

**Response to Reviewers**

We thank the reviewers for their detailed comments and helpful suggestions. We have addressed each comment below, with the Referee comment in **bold italicized text**, our response in plain text, and any manuscript changes noted in blue text. In addition, the revised manuscript with changes marked up has been attached to the end of our response to reviewers.

*Reviewer # 1*

*Comments:*
***Zhang et al. present a new experimental setup measuring glass transition temperature (Tg) using dielectric spectroscopy of thin films. Aerosol particles are deposited on an interdigitated electrode device using an electrostatic precipitator, eventually forming a thin film. The authors propose a new data analysis approach using the broadband dielectric spectroscopy data to determine Tg. The advantage of the technique compared to more traditional calorimetric techniques is that it needs considerably less material, hence it may open the possibility to measure ambient aerosol. Clearly, that makes the paper important and well suited to be published in AMT. The authors use two surrogates for secondary organic aerosol (SOA) to demonstrate that their technique yields glass transition temperatures comparable to reference methods. Both are pure compounds (glycerol and citric acid), no simple mixture nor an aqueous solution was investigated.***

***However, as one of the two tests (citric acid) show quite substantial deviation in measured Tg (307 K) compared to what is measured with calorimetric techniques (281- 285 K) -when the data are analyzed using the new technique - there is little evidence (namely, only the glycerol data) that the technique will actually work for ambient or laboratory SOA particles. As there are ample data available for pure compound SOA surrogates, aqueous SOA surrogates, as well as for simple mixtures [e.g. Lienhard et al., 2012; Dette et al., 2015; Dette and Koop, 2015], I think it is essential to perform measurements on more pure component surrogates, at least one mixture and – if possible – also an aqueous surrogate to characterize the experimental setup. As it is now, the paper does not allow the reader to judge whether the technique is actually feasible or needs further improvements before being applied to samples of unknown Tg.***

We thank the reviewer for reading and reviewing our manuscript, as well as his/her feedback. The reviewer brings up an excellent point of why we chose to use glycerol in this study. Glycerol is commonly used as a compound for characterizing and calibrating the performance of the dielectric spectroscopy. If the dielectric results of glycerol at all temperatures and all frequencies match the previous literature, then the dielectric spectrometer is considered calibrated and all other compounds that have dielectric spectra should show similar results. As shown in Figure 3, the dielectric measurements of glycerol in our studies match the previous studies very well, showing that the dielectric spectrometer in our study is accurate and is able to give accurate results. We also added a new plot comparing the data from the literature with this work at several different temperatures, as shown in Figure S1. The results match very well. We added the following text to further validate this method.

"The comparison of the dielectric relaxation timescale of glycerol measured from this study with literature values is shown in Figure S1, for several different temperatures. The results show that our measurements match the previously published results in the super-Arrhenius region when the compound is in equilibrium at a given temperature, with almost identical values. As the temperature continues to drop, glycerol and other compounds we tested fall out of equilibrium and become glass, which exhibits Arrhenius behavior. The transition from super-Arrhenius to Arrhenius behavior in this study provides the kinetically controlled glass transition as the compounds change from liquid to glassy state."

We understand the reviewer's interests to see more results to further validate this method. We acknowledged the work by previous publications (Dette et al., 2014; Lienhard et al., 2014; Dette and Koop, 2015) in the manuscript and examined three additional organic aerosols that have known glass transition temperatures, and the results from this study agree with literature values. We added text and plots in the main manuscript to include the above results.

"Glycerol, 1,2,6-hexanetriol, di-n-butyl phthalate, dioctyl phthalate, and citric acid particles were generated through the homogeneous nucleation method and atomizer method, respectively. The measured dielectric spectra of each compound show distinct relationships of their dielectric relaxation timescales with a 5 K/min cooling rate, as shown in Figure 5.

[Figure]

Figure 5. A plot of superimposed data points and curves constructed for glycerol, 1,2,6-hexanetriol, di-n-butyl phthalate, and dioctyl phthalate cooled at 5K/min. The solid color lines represent the fitted curves for the super-Arrhenius and Arrhenius region. The intersection between the two lines indicates the kinetically controlled glass transition region for each compound. The glass transition at a 5K/min cooling rate for each compound is shown in the plot."

The focus of this manuscript is to show that the thin film technique coupled with the dielectric spectroscopy is able to provide kinetically controlled glass transition results for submicron organic aerosol particles, and that cooling rates can affect the glass transition. However, whether this technique is fully applicable to all types of mixtures, aqueous aerosol systems, and even ambient aerosols, is currently being investigated by our team and is not within the scope of this manuscript. We added the following text to illustrate the future directions for this study. "Future research needs to be performed to examine organic mixtures and even secondary organic aerosols (SOA) using this method."

***Detailed comments:***

***1 Introduction:***

***I suggest add a paragraph explaining that one important property needed to understand kinetic limitations in gaseous uptake or loss of compounds in atmospheric aerosol particles is the diffusivity of this compound in the condensed phase. Measuring Tg of ambient SOA will indicate that below/at this temperature kinetic limitations will occur, but it does not give immediate quantitative insight.***

We thank the reviewer for this comment. The reason why we did not add this part earlier, just as the reviewer pointed out, was due to the fact that the glassy state is not a good guideline for kinetic limitations, as particles in certain semi-solid states can also impose kinetic limitations on the uptake and multiphase reaction of gas phase species. But it is an important motivation for studying phase states, and therefore we added new text to explain the effect of the glassy phase state on kinetic limitations.

"The phase state of aerosol particles also influences the diffusion of the gas phase species into the atmosphere, affecting the oxidation extent and multiphase reactions of the particles. For example, Shiraiwa and Seinfeld (2012) used models to predict that when aerosol particles are in certain semi-solid and glassy phase states, the reactive uptake of gas phase species will be kinetically limited. Kuwata and Martin (2012) showed that the phase state of secondary organic aerosols (SOA) affects the uptake of ammonia into the particles. Zhang et al. (2018) provided experimental and modeling evidence that the reactive uptake of isoprene-derived epoxydiols (IEPOX) into acidic sulfate particles is influenced by the phase state, which can contribute to at least a 30% reduction of isoprene-derived SOA in the Southeast U.S."

***Technical comment: Page 3, line 60: "nucleate" seems not the right wording here, better use "become activated" or similar.***

We thank the reviewer for this comment. "Nucleate" has been changed to "have liquid water condense on them"

***2 Experimental Setup:***

***One information missing is how much aerosol mass is needed for producing the film on the device. At least the area of the device and the approximate film thickness should be given. Even better, if the authors can provide the mass concentration in the aerosol flow prior to the precipitator and the time needed to accumulate for forming the film on the device.***

We added the following text in the manuscript to address the reviewer's comment. "The volume concentration of the aerosol particles at the inlet of the precipitator was $4.5 \times 10^{11}$ $nm^3$ $cm^{-3}$. After collecting for 5 hours, the film thickness is estimated to be 1-2 μm on 1 mm × 8 mm substrate based on the difference of volume concentration between the inlet and the outlets of the precipitator, the flow rate, the collection time, and assuming 50% collection efficiency.

*3 Data analysis:*

*I do not see the advantage of not adding the information of the SI into the main text. Please. Incorporate it including the raw data of Fig. S1 into section 3.1.*

We moved the Figure S1 to section 3.1 and named it Figure 6. We also added the following text in the main text to call out Figure 6.

"The kinetically controlled dielectric spectra fitted using Eq. (S2) are shown in Figure 6. From the fitting results in Figure 6, the dielectric relaxation timescales τ are derived as a function of temperature, as shown in Figure 7.

[Figure]

**Figure 6**. The dielectric relaxation spectrum of citric acid at different temperatures. The solid circles are measured experimental data and the solid lines are fitting curves parameterized from Eq. (S2) and Adrjanowicz et al. (2009)."

*I also find section 3.2 difficult to read as a non-expert. How does the discussion here connect to the issues of determining "fragility" [e.g. Angell, 2002]? I am not convinced that the glass community agrees that the intercept shown in Fig. 4 is the "true" glass transition.*

We thank the reviewer for the comment. Fragility, as described in Angell et al. (2002), is a parameter used to characterize the steepness of the slow down of the dynamics of glass-forming liquids as temperature is decreased. In this paper, we use dynamical facilitation theory as the scientific basis of the study, which is known to capture the behavior of various liquids over a large range of fragilities. According to this theory, the analog of the steepness, i.e., the curvature of the log relaxation time versus inverse temperature curve, is a parameter J, which is described in section 3.2 of the main text. J is unique to each material and can be related to the rate of motion of individual molecules (Keys et al., 2013). We added the following text to explain the theory in detail.

"The method used in our studies is based on dynamical facilitation theory (Elmatad et al., 2009; Chandler and Garrahan, 2010; Keys et al., 2011; Keys et al., 2013; Hudson, 2015, Hudson and Mandadapu, 2018), which also takes into account the effect of cooling rate on the glass transition. According to this theory, as a compound is cooled and transitions from a liquid to a supercooled liquid it exhibits super-Arrhenius behavior given by the following equation:

$$\log \tau/\tau_0 = J^2(1/T - 1/T_o)^2 \qquad (3)$$

where J is an energy scale intrinsic to each material related to the rate of motion of individual molecules, $T_o$ is termed the "onset temperature" and refers to the temperature at which a liquid showing Arrhenius relaxation becomes a supercooled liquid showing super-Arrhenius relaxation, and $\tau_0$ is a temperature-independent reference time scale of the order of the time taken for molecules to locally rearrange (Keys et al., 2011)"

Further, the idea of using the intersection between the super-Arrhenius and Arrhenius regions to determine glass transition temperature as a compound is cooled or warmed has been discussed and supported by publications from various research groups (Saiter et al., 2007; Shinyashiki et al., 2008). The predictions of the physical chemistry theory behind this idea, i.e., dynamical facilitation theory, agrees well with laboratory experiments (Keys et al., 2013).

The traditional method used $\tau$=100 s as the glass transition temperature, because the experiments were performed at equilibrium, i.e., the temperature of the chamber is set to one value, and then changed to the next one. Because those compounds are always measured at equilibrium, it is difficult to observe the transition from super-Arrhenius to Arrhenius regimes. Recently studies have shown that $\tau$ need not always be 100 s for all liquids (Saiter et al., 2007; Bahous et al., 2014) and the dielectric spectroscopy community does mostly agree that intersection between the two regimes reflects the kinetically controlled $T_g$ more accurately (Saiter et al., 2007; Shinyashiki et al., 2008). Therefore we respectfully disagree with the reviewer: the intersection between the two regimes as an indication of the "true" glass transition is one of the methods used in the literature to determine the value of the glass transition. We use this technique to determine the glass transition temperature because it has a theoretical basis in dynamical facilitation theory, and does not involve an empirical or arbitrary reference point such as $\tau$=100 s for the determination of the glass transition.

*Fig. 4: The caption needs to say which material is shown. Or, is this just a sketch to show the idea? Instead of "given cooling rate" in the legend, the actual cooling rate should be provided. They are no error bars shown for the data points: Are these actually the measurements, or some points arbitrarily taken from the fits? You need to provide more detail here.*

Figure 4 is an arbitrary plot for illustration purposes, not real measurement. To clarify, we have added the following text:

"Figure 4 shows a typical relationship between the dielectric relaxation timescale and the temperature, as a compound is cooled down or warmed up between the liquid state and glassy state."

We also changed the caption for Figure 4:

"Figure 4. An illustrated plot of the relationship between dielectric relaxation time scale and temperature. The logarithm of the relaxation is plotted against inverse T, i.e., log $\tau$ vs 1000/T. By linking the data points together, one can plot the super-Arrhenius curve (red) and the Arrhenius line (green). Using a consistent cooling rate, the intersection of the two regions identifies the compound's glass transition temperature, as indicated by the shaded blue region. The intersection of the two black lines represents the glass transition point. The intersection of the two black dashed lines shows the glass transition temperature determined using traditional method when $\tau$=100 s."

*4 Results and discussion:*

***First, there is more data available in the literature to compare your data with: for glycerol see for example Lienhard et al. (2012). Citric acid has been measured by Lienhard et al. [2012] as well and by Dette et al. [2014].***

We thank the reviewer for the comment. We have included the references that the reviewer provided. However, we would like to point out that these studies use calorimetry for their measurements in contrast with our dielectric relaxation spectroscopy measurements. Studies have shown that the glass transition temperature can change significantly between these techniques, by 5-10 K or sometimes even up to 50 K (Angell, 2002; Shinyashiki et al., 2008), making comparison difficult.

***Second, in Fig. 5 you determined Tg of citric acid as 305-315 K, whereas in Table 1 you write 307+-5 K. Please check.***

We thank the reviewer for the comment. We have examined the data and updated it to be 302-312K, as shown below:

[Figure]

**Figure 7.** A plot of superimposed data-points and curves constructed for citric acid warmed at 5K/min. The solid color lines represent the fitted curves for the super-Arrhenius and Arrhenius region. The blue shaded area shows the glass transition region. The two vertical black lines associated with the blue shaded area indicate the corresponding temperature range where the super Arrhenius curve intersects with the Arrhenius line. The traditional glass transition temperature, i.e., the temperature when $\tau$=100 s, is also marked.

*Obviously, Fig. 5 shows a very significant difference when comparing the "classical" determination (tau = 100s) for Tg (281 K) and your new method (305-315 K). As the former agrees with the calorimetric measurements of numerous experiments in the literature, whereas the latter show significant deviation, you definitely need more evidence than provided to support the new method. At present, the reader will conclude that the new method is not reliable.*

We thank the reviewer for the comment. We understand the reviewer's concern. We would like to respectfully point out that using the intercept to determine the glass transition is not a "new" method. The theory and related discussions of its advantages have been published previously by multiple research groups (Keys et al., 2013; Bahous et al., 2014; Limmer and Chandler, 2014; Hudson, 2015). Within the dielectric relaxation method, the difference between using $\tau=100$ s and the intercept method is that one measures the equilibrium state between the compound and the temperature, while the latter measures the dynamic process with various cooling or heating rates (Saiter et al., 2007; Bahous et al., 2014).

We would also like to point out the DSC method and the dielectric method do not have to be in exact agreement. Due to the differences in measurement parameters and definitions of the glass transition, different measurements can give different Tg values (Shinyashiki et al., 2008). DSC uses heat capacity to measure the glass transition while dielectric relaxation uses molecular variations to define the glass transition. Angell (2012) also pointed out in his paper that there are at least three different definitions of the glass transition and the Tg determined by each can be 50 K different from each other.

To include more evidence, we now show data for three other compounds besides glycerol and citric acid, and the results show very good agreement with the previous studies, not just for Tg measurement (as shown in Table 1, which could depend on the definition used), but also for the dielectric spectra at each temperature (Figure S1 in the SI and Figure 5 in the main text). The additional results further validate the results from our method. Please see the revised text, table, and figure below with additional data.

"As is shown in Table 1, the kinetically controlled glass transition data agree well with previously measured literature values of glycerol (Zondervan et al., 2007; Chen et al., 2012; Amann-Winkel et al., 2013), 1,2,6-hexanetriol (Nakanishi and Nozaki, 2010), di-n-butyl phthalate (Dufour et al., 1994), dioctyl phthalate (Beirnes Kimberley and Burns Charles, 2003), within 3% of the literature value. The comparison of the dielectric relaxation timescale of glycerol measured from this study with literature values is shown in Figure S1, for several different temperatures. The results show that our measurements match the previously published results in the super-Arrhenius region when the compound is in equilibrium with the temperature, with almost identical values.

**Table 1.** Glass transition temperatures for selected organic species at selected cooling or heating rates measured by broadband dielectric spectroscopy with a thin-film interdigitated electrode array

| Compound | Chemical Formula | $T_g$ (K)-Measured | $T_g$ (K)-Literature |
|---|---|---|---|
| Glycerol | $C_3H_8O_3$ | <189 K (2 K/min) | 190 K (Zondervan et al., 2007) |
| | | $192 \pm 2$ K (5 K/min) | 191 K (Chen et al., 2012) |
| | | $194 \pm 2$ K (10 K/min) | 196 K (Amann-Winkel et al., 2013) |
| | | | $191.7 \pm 0.9$ K (Lienhard et al., 2012) |
| 1,2,6-hexanetriol | $C_6H_{14}O_3$ | $192 \pm 2$ K (5 K/min) | 196 K (Nakanishi and Nozaki, 2010) |
| di-n-butyl phthalate | $C_{16}H_{22}O_4$ | $180 \pm 2$ K (5 K/min) | 174 K (Dufour et al., 1994) |
| dioctyl phthalate | $C_{24}H_{38}O_4$ | $194 \pm 2$ K (5 K/min) | 190 K (Beirnes Kimberley and Burns Charles, 2003) |
| Citric Acid | $C_6H_8O_7$ | $307 \pm 5$ K (5 K/min) | $281 \pm 5$ K (Bodsworth et al., 2010)[*] |
| | | | $285 \pm 0.2$ K (Lu and Zografi, 1997) |
| | | | $281.9 \pm 0.9$ K (Lienhard et al., 2012) |
| | | | 283-286 K (Dette et al., 2014) |
| | | | $260 \pm 10$ K (Murray, 2008)[**] |

[*] The data was based on modeling result.  [**]The data was based on fitting extrapolation result.

[Figure]

**Figure S1**. The logarithm of the relaxation timescale as a function of the inverse temperature derived from glycerol. The circles are from Lunkenheimer et al. (1999) and Angell (1995). The triangular points are experimental measurements from this work with different cooling rates. The dashed lines are the fitted curves for the super-Arrhenius and Arrhenius regions. The results show that the Tg values from this work match very well with previous studies."

For citric acid, comparison of both the dielectric spectra as well as the derived Tg with literature values using the same method is difficult because there have not been any other dielectric relaxation experiments on citric acid published to our knowledge. The studies we have compared with are calorimetry measurements. It has been shown that the glass transition temperature can vary significantly depending on the experimental technique used, as well as the cooling or heating rates (Angell, 2002). Furthermore, the data shown for citric acid were obtained during a heating protocol, which can be significantly different from data obtained during a cooling protocol due to hysteresis. Finally, it is likely that citric acid is a less fragile liquid, i.e., the glass transition temperature depends strongly on cooling or heating rates. One example of such a compound is borosilicate. Moynihan et al. (1974) show that a change of 2 K/min to 10 K/min cooling rate could change the glass transition temperature of borosilicate glass by 15 K.

Since we have found no publications presenting the dielectric spectra of citric acid, we believe that our results are worth publishing here. However, we include the following text in section 4.1 to discuss the difference in the measurements and the literature.

" (1) To date there have been no measurements of the dielectric spectra for citric acid to our knowledge. The references in Table 1 are based on DSC measurements, which may not be in exact agreement with the dielectric measurement. Angell (2012) pointed out that there are at least three different definitions of the glass transition and the Tg determined by each can be 50 K different from each other. DSC uses heat capacity change to measure the glass transition while dielectric relaxation uses molecular movements to define the glass transition. Due to the difference in measurement parameters and the definition of the glass transition, these two measurements can provide different Tg values of the same compound by up to 10 K (Shinyashiki et al., 2008). (2) This study focuses the kinetically controlled glass transition temperature at a given cooling rate, i.e., the transition of the long range intermolecular movements of the sample to determine the glass transition temperature. If the more conventional method, i.e. fitting the super-Arrhenius curve to obtain the glass transition temperature in when $\tau$=100 s, is applied to the data, as shown in Figure 7, the obtained glass transition temperature would be $281 \pm 3$ K, which is within 1% difference compared with the two nearest literature values. However, as previous publications have pointed out (Keys et al., 2013; Bahous et al., 2014; Limmer and Chandler, 2014; Hudson, 2018), using the transition from super-Arrhenius to Arrhenius region is likely to reflect the true glass transition when taking kinetic factors, such as cooling/heating rates, into consideration. (3) Moreover, glass transition data of citric acid are rather limited and there are differences between each study. For instance, from the literature data, the differences between three reported glass transition temperatures are up to 10%, which is the same value comparing our data to the other two nearest literature results. (4) The heating and cooling rates may also contribute to the difference of the $T_g$ of citric acid between the literature and this study, as this study uses a lower cooling rate than the ones reported by the literature. Moynihan et al. (1974) reported that a change of 2 K/min to 10 K/min cooling rate could alter the glass transition temperature of borosilicate by 15 K. It is possible that citric acid is a less fragile liquid, similar to borosilicate, i.e., the glass transition temperature depends strongly on cooling or heating rates."

***Concerning the influence of cooling rates, I recommend to the authors to make use of the detailed study of Simatos et al. [1996], do an additional experiment using sorbitol or fructose with the setup, and compare the results with those of Simatos et al. [1996].***

We thank the reviewer for pointing out the Simatos et al. paper. We have included it in our manuscript. Simatos et al. shows that Tg changes about 3-5K as the cooling rate changes from 2 K/min to 10 K/min. Unfortunately, sorbitol or fructose have relatively small dipole moments, making it very difficult to perform dielectric analysis for our current instrumental sensitivity. However, the change of 3-5 K reported in Simatos et al. agrees well with our study. Herein, we included the text to discuss this agreement reported by Simatos et al.

"This study reports an increase of 5 K or more in the glass transition of glycerol as the cooling rate changes from 2 K/min to 10 K/min. The reported increase in glass transition temperature during these cooling rates agrees with the behavior of sorbitol and fructose in another study performed by Simatos et al. (1996), which also measured the dependence of glass transition temperature on cooling rate with small organic molecules."

*Comments:*
*This study developed a method to measure the kinetically-controlled glass transition temperatures using the broadband dielectric spectroscopy. Aerosol particles are de- posited in the form of a thin film on an electrode using electrostatic precipitation. Glass transition temperatures of glycerol and citric acid were measured, agreeing reason- ably well with available literature data. By considering the intersect of super Arrhenius and Arrhenius lines, the new method appears to provide more reliable glass transition temperatures. The effect of cooling rates on glass transition temperatures were also investigated. I found that the measurements were conducted in a highly elegant way. The manuscript is clearly written and easy to follow. In fact, I enjoyed reading this manuscript very much.*

*My only major concern is that the method was validated with only two compounds. The authors plan to apply this method to measure glass transition temperatures of highly complex SOA mixtures in follow-up studies (L342). In this sense, it would be essential to test a few more compounds. There are indeed a number of organic compounds with known Tg, and I wonder why the authors would not have conducted some more measurements to consolidate this new method. Is this method too time-consuming or not so easy to be applied with more compounds?*

We thank the reviewer for reading and reviewing our manuscript and his/her feedback. The reviewer brings up having more data to verify this technique, which we agree and did include more evidence. We would like to explain why we chose to use only glycerol in this study first. Glycerol is commonly used as a compound for characterizing and calibrating the performance of the dielectric spectroscopy. If the dielectric results of glycerol at all temperatures and all frequencies match previous literature, then the dielectric spectrometer is considered calibrated and all other compounds that have dielectric spectra should show similar results. As shown in Figure 3, the dielectric measurements of glycerol in our studies match the previous studies very well, showing that the dielectric spectrometer in our study is accurate and is able to give accurate results. We added Figure S1 to further prove that the derived relaxation timescales from our study also match the values from other literature at several different temperatures for glycerol.

We understand the reviewer's interest to see more results to validate the method. To include more evidence, we now show data for three other compounds besides glycerol and citric acid, and the results show very good agreement with the previous studies, not just for Tg measurement (as shown in Table 1, which could depend on the definition used), but also for the dielectric spectra at each temperature (Figure 5 in the main text and Figure S1 in the SI). The additional results further validate the results from our method. Please see the revised text, table, and figure below with additional data.

"As is shown in Table 1, the kinetically controlled glass transition data agree well with previously measured literature values of glycerol (Zondervan et al., 2007; Chen et al., 2012; Amann-Winkel et al., 2013), 1,2,6-hexanetriol (Nakanishi and Nozaki, 2010), di-n-butyl phthalate (Dufour et al., 1994), dioctyl phthalate (Beirnes Kimberley et al., 2003), and are within

3% of the literature value. The comparison of the dielectric relaxation timescale of glycerol measured from this study with literature values is shown in Figure S1, for several different temperatures. The results show that our measurements match the previously published results in the super-Arrhenius region when the compound is in equilibrium at a given temperature, with almost identical values.

**Table 1.** Glass transition temperatures for selected organic species at selected cooling or heating rates measured by broadband dielectric spectroscopy with a thin-film interdigitated electrode array

| Compound | Chemical Formula | $T_g$ (K)-Measured | $T_g$ (K)-Literature |
|---|---|---|---|
| Glycerol | $C_3H_8O_3$ | <189 K (2 K/min) | 190 K (Zondervan et al., 2007) |
| | | 192 ± 2 K (5 K/min) | 191 K (Chen et al., 2012) |
| | | 194 ± 2 K (10 K/min) | 196 K (Amann-Winkel et al., 2013) 191.7 ± 0.9 K (Lienhard et al., 2012) |
| 1,2,6-hexanetriol | $C_6H_{14}O_3$ | 192 ± 2 K (5 K/min) | 196 K (Nakanishi and Nozaki, 2010) |
| di-n-butyl phthalate | $C_{16}H_{22}O_4$ | 180 ± 2 K (5 K/min) | 174 K (Dufour et al., 1994) |
| dioctyl phthalate | $C_{24}H_{38}O_4$ | 194 ± 2 K (5 K/min) | 190 K (Beirnes Kimberley and Burns Charles, 2003) |
| Citric Acid | $C_6H_8O_7$ | 307 ± 5 K (5 K/min) | 281 ± 5 K (Bodsworth et al., 2010)[*] 285 ± 0.2 K (Lu and Zografi, 1997) 281.9 ± 0.9 K (Lienhard et al., 2012) 283-286 K (Dette et al., 2014) 260 ± 10 K (Murray, 2008)[**] |

[*] The data was based on modeling result. [**]The data was based on fitting extrapolation result.

[Figure]

**Figure S1**. The logarithm of the relaxation timescale as a function of the inverse temperature derived from glycerol. The circles are from Lunkenheimer et al. (1999) and Angell (1995). The triangular points are experimental measurements from this work with different cooling rates. The dashed lines are the fitted curves for the super-Arrhenius and Arrhenius regions. The results show that the Tg values from this work match very well with previous studies."

*Minor comment: - L61: I suggest citing also Zobrist et al., PCCP, 13, 3514, 2011.*

We thank the reviewer for the comment. Yes we agree this paper is very relevant and we have included the references that the reviewer provided.

*- L72: Rothfuss and Petters, PCCP, 2017 measured the viscosity of sucrose but not various types of SOA (or did they?).*

We thank the reviewer for the comment. The sentence has been changed to: "Rothfuss and Petters (2017) studied the viscosities of sucrose particles with sodium dodecyl sulfate (SDS) particles up to $10^7$ Pa under sub-freezing temperature regimes."

*- L282: "the" should be replaced to "that".*

The word has been corrected.

*- SI contains useful information (especially Figure S1 is interesting) and I suggest moving them into the main manuscript or appendix.*

Thank you for the suggestion. We moved the Figure S1 to section 3.1 and and named it Figure 6. We also added text to introduce Figure 6 as shown below.

"Our kinetically controlled dielectric spectra fitted using Eq. (S2) is shown in Figure 6. From the fitting results in Figure 6, the dielectric relaxation timescales $\tau$ are derived as a function of temperature, as shown in Figure 7.

[Figure]

**Figure 6**. The dielectric relaxation spectrum of citric acid at different temperatures. The solid circles are measurement experimental data and the solid lines are fitting curves parameterized from Eq. (S2) and Adrjanowicz et al. (2009). "

.

[revised manuscript text omitted]

---

## Author Response (AR2)

**Response to Reviewers**

We thank the reviewers for their detailed comments and helpful suggestions. We have addressed each comment below, with the Referee comment in **bold italicized text**, our response in plain text, and any manuscript changes noted in blue text. In addition, the revised manuscript with changes marked up has been attached to the end of our response to the Referee.

**Reviewer # 1**

**Comments:**

My major criticism of the paper was that only two components were used to show the feasibility of the method. The authors have now measured three additional substances showing quite reasonable agreement with data available in the literature. It is interesting to note that DSC measurements of Glycerol and DOP agrees quite well, while citric acid shows a stronger deviation.

I recommend publishing the revised version after the correction of some technical issues.

In the updated Table 1: Nakanashi and Nozaki report as Tg for the hexanetriol (200+-2) K and not 196 K!

There are two additional data for this compound in the literature: (206.4+-0.5) K from Dorfmüller et al. (1979) and 202 K from Böhmer et al. (1993).

The citation of Beirnes and Burns in J. Appl. Polym. Sci. paper has a wrong publication date: it is 1986 and not 2003.

[revised manuscript text omitted]

| Compound                | Chemical
Formula                           | Tg (K)-Measured      | T g (K)-Literature                 |
|-------------------------|-----------------------------------------------|----------------------|-----------------------------------------------|
|                         | C 3 H 8 O 3  | <189 K (2 K/min)     | 190 K (Zondervan et al., 2007)                |
| Glycerol                |                                               | 192 ± 2 K (5 K/min)  | 191 K (Chen et al., 2012)                     |
| Giyeeloi                |                                               | 194 ± 2 K (10 K/min) | 196 K (Amann-Winkel et al., 2013)             |
|                         |                                               |                      | $191.7 \pm 0.9$ K (Lienhard et al., 2012)     |
|                         | C 6 H 14 O 3 |                      | $200 \pm 2$ K (Nakanishi and Nozaki,
2010) |
| 1,2,6-
Hovenstrial   |                                               | 192 ± 2 K (5 K/min)  | $206.4 \pm 0.5$ K (Dorfmüller et al.,         |
| TTEXAILEUTOI            |                                               |                      | 1979)                                         |
|                         |                                               |                      | 202 K (Böhmer et al., 1993)                   |
| Di-n-butyl
Phthalate | $C_{16}H_{22}O_4$                             | 180 ± 2 K (5 K/min)  | 174 K (Dufour et al., 1994)                   |
| Dioctyl
Phthalate    | $C_{24}H_{38}O_4$                             | 194 ± 2 K (5 K/min)  | 190 K (Beirnes Kimberley et al.,
1986)     |
|                         | C 6 H 8 O 7  | 307 ± 5 K (5 K/min)  | 281 + 5 K (Bodsworth et al 2010)*             |
|                         |                                               |                      | $285 \pm 0.2$ K (Lu and Zografi, 1997)        |
| Citric Acid             |                                               |                      | $281.9 \pm 0.9$ K (Lienhard et al., 2012)     |
|                         |                                               |                      | 283-286 K (Dette et al., 2014)                |
|                         |                                               |                      | $260 \pm 10$ K (Murray, 2008) **   |

dielectric spectroscopy with a thin-film interdigitated electrode array

\* The data was based on modeling result. \*\*The data was based on extrapolation of a fit to the data.

**List of Figures**

**Figure 1.** A schematic diagram of the experimental setup and procedure. The experimental approach consists of four parts: aerosol generation, thin film deposition using an electrostatic precipitator, the temperature control system, and the broadband dielectric spectroscopy measurement system.